# Random Rank-Dependent Expected Utility

**Nail Kashaev †** 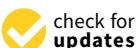 **and Victor H. Aguiar ***

Department of Economics, University of Western Ontario, London, ON N6A 3K7, Canada; nkashaev@uwo.ca
* Correspondence: vaguiar@uwo.ca
† The symbol indicates that the authors' names are in certified random order.

**Abstract:** We present a novel characterization of random rank-dependent expected utility for finite datasets and finite prizes. As a byproduct, we obtain a characterization of random expected utility that works for finite datasets. The test lends itself to statistical testing. We apply our test to an experimental dataset and find evidence against random expected utility, while random rank-dependent expected utility can explain the dataset.

**Keywords:** random utility; expected utility; rank-dependent expected utility



## 1. Introduction

The rank-dependent expected utility (RDEU) model, first proposed to study decision under risk by Quiggin [1], is one of the main alternatives to the expected utility (EU) model. (Schmeidler [2] characterized RDEU to study decision under uncertainty. Abdellaoui [3] provided a characterization of RDEU for risk.) RDEU is popular because it extends EU to accommodate known empirical anomalies that violate the predictions of EU (e.g., Allais Paradox), while at the same time, it produces sharp comparative statics and preserves transitivity [4]. It was also used to extend Prospect Theory [5] to Cumulative Prospect Theory [6]. (See Diecidue and Wakker [7] and Quiggin [8] for a review and references therein.)

The RDEU model extends EU by allowing cumulative probabilities of lotteries to be weighted by a weighting function. This means that RDEU does not rely on the independence axiom. (The independence axiom dictates that for two lotteries $p$ and $q$, $p$ is weakly preferred to $q$ if and only if $\alpha p + (1 - \alpha)r$ is weakly preferred to $\alpha q + (1 - \alpha)r$ for any lottery $r$, and $alpha \in [0,1]$.) According to Quiggin [4], RDEU is EU with respect to a transformed probability distribution. Formally, we say a decision maker (DM) that can be described by a RDEU ranks lottery $p$ over lottery $q$ if and only if there exists a (Bernoulli) utility function $u$ and a weighting function $\phi : [0,1] \to [0,1]$, $\phi(0) = 0$, $\phi(1) = 1$, such that

$$\sum_{k=1}^{K} \left[ \phi\left(\sum_{t=1}^{k} p_t\right) - \phi\left(\sum_{t=1}^{k-1} p_t\right) \right] u(x_k) > \sum_{k=1}^{K} \left[ \phi\left(\sum_{t=1}^{k} q_t\right) - \phi\left(\sum_{t=1}^{k-1} q_t\right) \right] u(x_k),$$

where prizes $x_k$ are ranked by some primitive order, and we use convention that $\sum_{t=1}^{0} p_t = 0$.

Here we study the empirical implications of RDEU when we observe a cross-section of choices. These choices are from a finite collection of menus and lotteries and are made by DMs that follow the RDEU rule. Since we allow the primitives of RDEU to be heterogeneous among the DMs (e.g., random $\phi$ and random $u$), we call this model of population behavior Random Rank-Dependent Expected Utility (RRDEU). We fully characterize RRDEU and provide a statistical test of it using the tools in Kitamura and Stoye [9] (henceforth KS).

As a byproduct of our characterization of RRDEU in a finite stochastic choice dataset (limited stochastic dataset), we provide a characterization of random expected utility (REU). The characterization of REU is the first of its type for limited stochastic datasets. The

seminal work of Gul and Pesendorfer [10] (henceforth GP) characterizing REU for an infinite stochastic dataset may provide false positives when applied to a limited stochastic dataset. The two main axioms in the GP characterization are regularity and independence. Regularity requires that the probability of choosing a lottery weakly decreases as the number of alternatives in a menu grows. Independence in GP requires that if we expand a menu *A* by mixing all the lotteries inside it with an outside lottery *r*, then the probability of choice of the mixture lotteries in the new menu is the same as the probability of choice of the original ones. The problem with the characterization in GP is that if we have only three menus that are not ranked by the set inclusion relation and are not related by the mixture operations required by the independence assumption in GP, then the axioms are trivially satisfied. However, we will show that even with three menus such as the ones we have described, there are still empirical properties of REU that may fail, showing that the test in GP fails for limited stochastic choice dataset.

**Example 1** (Counterexample to [10]). *Consider three lotteries $\{p, q, p'\}$, where $p' = \alpha p + (1 - \alpha)r$ for some $\alpha \in (0, 1)$ and some lottery r. Assume that we observe choice probabilities from three menus: $\{p, q\}$, $\{p', q\}$, and $\{p', p\}$. These choice probabilities are such that $\rho_{\{p,q\}}(p) = 1$, $\rho_{\{p',p\}}(p') > 0$, and $\rho_{\{p',q\}}(p') = 0$, where $\rho_A(a)$ denotes the probability of choosing a in menu A. The axioms of independence and regularity of GP are trivially satisfied. Thus, if the DMs can be described by EU behavior, then a positive probability mass should be put on an expected utility function U such that $U(r) > U(p)$ (since $\rho_{\{p',p\}}(p') > 0$) and $U(p) > U(q)$ (since $\rho_{\{p,q\}}(p) = 1$). This is impossible, since in menu $\{p', q\}$, lottery p' is picked with zero probability ($\rho_{\{p',q\}}(p') = 0$). This means that this limited stochastic dataset is inconsistent with REU.*

Example 1 shows that the test in GP fails when we consider a limited stochastic dataset. Since EU is a special case of RDEU, this is also a problem for the latter. Indeed, similar issues will appear in GP's style characterizations of RRDEU. To fix ideas about the axiomatization of RDEU provided by Abdellaoui [3], the main axiom that replaces the independence axiom is the *probability trade-off consistency* condition. This condition requires a very rich domain of lotteries. This means that, generically, the probability trade-off consistency condition fails to have empirical bite in incomplete datasets such as that studied in Example 1. (Schmidt and Zank [11] simplifies the domain requirements in Abdellaoui [3], but the resulting domain is still rich and does not cover finite datasets. In addition, none of these characterizations deal with a cross-section of choices).

Our results provide a fix for this issue. The intuition behind our test is that for any linear order on the set of finite lotteries, we can uncover whether there is *u* and $\phi$ that can describe this order and, hence, make it consistent with RDEU. If the answer to the previous question is affirmative, then we can use that linear order as the support of a random utility rule. We show that this test can be done using a simple quadratic program in the case of RRDEU that simplifies to a linear program in the case of REU.

Since we have characterized the empirical content of RRDEU that has a natural population interpretation, our tests can be used in experimental and field datasets that have a cross-section of choices with limited menu variation. Finally, since our test lends itself to statistical testing, it can deal with sampling variability, which is not the case with GP. To the best of our knowledge, no other work has provided a characterization and statistical test for RRDEU in a cross-section of choices. The closest work to ours is Polisson et al. [12]. It provides a nonparametric test of EU and RDEU for a setup with linear budgets and with a time-series of choices from the same individual. We consider cross-sections.

In Section 2, we describe our setup and formally define RRDEU. In Section 3, we provide a construction of the set of all linear orders consistent with RDEU for a given finite set of lotteries and establish the main theoretical result. In Section 4, we apply our method to an experimental dataset. Finally, we conclude in Section 5.

## 2. Model

We consider a finite set of distinct alternatives $X = \{x_k\}_{k=1}^K \subset \mathbb{R}$ such that $x_k > x_{k+1}$ for all $k = 1, \ldots, K-1$. Let $\Delta(X)$ be the set of all lotteries (distributions) defined on $X$ and $\Pi \subseteq \Delta(X)$ be a finite subset of it. Let $\mathcal{A} \subseteq 2^{\Pi} \setminus \{\varnothing\}$ be a collection of menus of lotteries. The probabilistic choice rules are $\rho_A \in \Delta(A)$, where $A \in \mathcal{A}$ and $\Delta(A)$ is the simplex defined on menu $A$. A stochastic choice dataset is $\rho = (\rho_A)_{A \in \mathcal{A}}$. The set of all linear orders on $\Delta(X) \times \Delta(X)$ is denoted by $S$. The symbol $\mathbb{1}(t)$ is equal to 1 when $t$ is true and is equal to 0 otherwise.

**Definition 1** (Random Utility, RU). *We say that $\rho$ admits a random utility (RU) representation if there exists $\mu \in \Delta(S)$ such that*

$$\rho_A(a) = \sum_{\succ \in S} \mu(\succ) \mathbb{1}(a \succ b, \forall b \in A)$$

*for all $A \in \mathcal{A}$ and $a \in A$.*

Let $\mathcal{U}$ be the set of all utility functions from $X$ to $\mathbb{R}$, and $\mathcal{P}$ be the set of all weighting functions $\phi : [0,1] \to [0,1]$ such that $\phi(0) = 0$ and $\phi(1) = 1$. Every $u$ and $\phi$ together define a rank-dependent expected utility function over the set of lotteries $\Delta(X)$ as

$$U_{u,\phi}(p) = \sum_{k=1}^{K} \left[ \phi\left(\sum_{t=1}^{k} p_t\right) - \phi\left(\sum_{t=1}^{k-1} p_t\right) \right] u(x_k)$$

for any $p \in \Delta(X)$, where we use convention that $\sum_{t=1}^{0} p_t = 0$. The rank-dependent expected utility function coincides with the standard expected utility function if $\phi(x) = x$.

Similar to the case with the expected utility function, the rank-dependent expected utility function allows us to define preference orders that are consistent with rank-dependent expected utility.

**Definition 2.** *We say that the linear order $\succ \in S$ is rank-dependent expected utility linear order or $\succ \in R$ if there exist $u \in \mathcal{U}$ and $\phi \in \mathcal{P}$ such that for any $p, q \in \Delta(X)$*

$$p \succ q \iff U_{u,\phi}(p) > U_{u,\phi}(q).$$

Given the set of all rank-dependent expected utility orders we can define when observed data $\rho$ could have been generated by a heterogeneous population of DMs with rank-dependent expected utility orders.

**Definition 3** (Random Rank-Dependent Expected Utility, RRDEU). *We say that $\rho$ admits a random rank-dependent expected utility (RRDEU) representation if there exists $\mu \in \Delta(R)$ such that*

$$\rho_A(a) = \sum_{\succ \in R} \mu(\succ) \mathbb{1}(a \succ b, \forall b \in A)$$

*for all $A \in \mathcal{A}$ and $a \in A$.*

The definition of RRDEU allows us to test whether a given data point is consistent with it the same way we do it for RU. The only difference is that instead of working with all possible strict linear orders $S$, we need to compute the set $R$.

## 3. Construction of the Set of Rank-Dependent Expected Utility Linear Orders

Before we describe the general procedure for construction of $R$, we explain how the procedure works with EU.

### 3.1. Expected Utility Model

As we mentioned before, the standard expected utility model is the special case RRDEU with $\phi(x) = x$. Thus, to check whether a given linear order $\succ$ is expected utility linear order, we need to check whether there exists a utility function $u$ such that

$$p \succ q \iff \sum_{k=1}^{K} p_k u(x_k) > \sum_{k=1}^{K} q_k u(x_k)$$

for all $p, q \in \Delta(X)$. Note that, in the spirit of revealed preference inequalities, the latter is equivalent to requiring the existence of a set of reals $\{v_k\}_{k=1}^{K}$ such that

$$p \succ q \iff \sum_{k=1}^{K} p_k v_k > \sum_{k=1}^{K} q_k v_k.$$

Since the last inequality must hold for all possible $p$ and $q$, we end up having the following system of linear inequalities:

$$\sum_{k=1}^{K} (p_k - q_k) v_k > 0, \quad p, q \in \Delta(X).$$

If there are finitely many lotteries (i.e., we consider $p, q \in \Pi$, $|\Pi| < \infty$), then let $p^{\succ(l)}$ denote the $l$-th best lottery in $\Pi$ according to $\succ$. In this case, it suffices to conduct $|\Pi| - 1$ comparisons to get the following system of linear inequalities:

$$\sum_{k=1}^{K} (p_k^{\succ(l)} - p_k^{\succ(l+1)}) v_k > 0, \quad l = 1, \ldots, |\Pi| - 1.$$

Checking whether a finite set of linear inequalities has a solution is a linear programming problem and can be done very efficiently and fast.

### 3.2. Rank-Dependent Expected Utility

Next, we describe how to extend the procedure for the standard expected utility model to the rank-dependent one. Given a candidate linear order $\succ \in S$, we need to check whether there exist $u$ and $\phi$ such that for any lotteries $p$ and $q$

$$\sum_{k=1}^{K} \left[ \phi\left( \sum_{t=1}^{k} p_t \right) - \phi\left( \sum_{t=1}^{k-1} p_t \right) - \phi\left( \sum_{t=1}^{k} q_t \right) + \phi\left( \sum_{t=1}^{k-1} q_t \right) \right] u(x_k) > 0.$$

For a finite set of lotteries, we can reformulate the problem as requiring existence of two sets of reals $\{v_k\}_{k=1}^{K}$ and $\{f_{l,k}\}_{k=1, l=1}^{K-1, |\Pi|}$ such that

$$\sum_{k=1}^{K} [f_{l,k} - f_{l,k-1} - f_{l+1,k} + f_{l+1,k-1}] v_k > 0, \quad l = 1, \ldots, |\Pi| - 1,$$

$$f_{l,k} = f_{s,m}, \text{ whenever } \sum_{t=1}^{k} p_t^{\succ(l)} = \sum_{t=1}^{m} p_t^{\succ(s)},$$

$$f_{l,k} = 0, \text{ whenever } \sum_{t=1}^{k} p_t^{\succ(l)} = 0, \tag{1}$$

$$f_{l,k} = 1, \text{ whenever } \sum_{t=1}^{k} p_t^{\succ(l)} = 1,$$

$$0 \leq f_{l,k} \leq 1.$$

The first set of inequalities just implies that the rank-dependent utilities lead to the order consistent with $\succ$. The rest of the constraints follow from the definition of $\phi$ (i.e., $\phi : [0,1] \rightarrow [0,1]$, $\phi(0) = 0$, and $\phi(1) = 1$). Checking whether system (1) is satisfied for some $\{v_k\}_{k=1}^{K}$ and $\{f_{l,k}\}_{k=1, l=1}^{K-1, |\Pi|}$ is a quadratic problem, which also can be solved efficiently and fast.

The next lemma formally establishes that system (1) provides necessary conditions for $\succ$ being a rank-dependent expected utility order.

**Lemma 1.** *Given $\Pi \subseteq \Delta(X)$, $|\Pi| < \infty$, the linear order $\succ \in S$ belongs to $R$ only if there exist $\{v_k\}_{k=1}^{K}$ and $\{f_{l,k}\}_{k=1, l=1}^{K-1, |\Pi|}$ such that system (1) is satisfied.*

**Proof.** The result trivially follows if one takes $v_k = u(x_k)$ and $f_{l,k} = \phi\left(\sum_{t=1}^{k} p_t^{\succ(l)}\right)$. $\square$

Next, we state the extension proposition that guaranties that system (1) leads to a rank-dependent expected utility order.

**Proposition 1.** *Given $\Pi \subseteq \Delta(X)$, $|\Pi| < \infty$, if there exist $\{v_k\}_{k=1}^{K}$ and $\{f_{l,k}\}_{k=1, l=1}^{K-1, |\Pi|}$ such that system (1) is satisfied for some $\succ^* \in \Pi \times \Pi$, then there exists $\succ \in R$ that coincides with $\succ^*$ on $\Pi \times \Pi$.*

**Proof.** The proposition requires building functions $u$ and $\phi$ from $\{v_k\}_{k=1}^{K}$ and $\{f_{l,k}\}_{k=1, l=1}^{K-1, |\Pi|}$. The simplest utility function that makes $\succ$ rank-dependent expected utility order is any piecewise linear function with nods at points $\{(x_k, v_k)\}_{k=1}^{K}$. To construct $\phi$, one can also take any piecewise linear function with nods at points $\{(\sum_{t=1}^{k} p^{\succ(l)}, f_{k,l})\}_{k=1, l=1}^{K-1, |\Pi|}$. $\square$

After the set $R$ is constructed, the problem of testing whether a stochastic dataset $\rho$ admits an RRDEU representation can be done by testing the restricted RU model as in McFadden and Richter [13] and KS.

Let $R_\Pi$ be the set of all linear orders on $\Pi \times \Pi$ that are consistent with RDE (i.e., the restriction of $R$ to $\Pi$). (The set of linear orders $R_\Pi$ can be replaced by the set of expected utility ranking). We can use system (1) to uncover the elements of $R_\Pi$. We want to test whether $\rho$ admits an RRDEU representation; this turns out to be equivalent to testing whether $\rho$ can be generated by a population of DMs whose preferences are in $R_\Pi$. Let $B$ be the matrix of the size $d_\rho \times |R_\Pi|$, where $d_\rho$ is the dimensionality of $\rho$, such that $(k, l)$ element of it is equal to

$$B_{k,l} = \mathbb{1}(a \in A)\mathbb{1}(a \succ_l c, \forall c \in A),$$

where $k$ corresponds to a pair $(a, A)$ such that $a \in A$, and $\succ_l$ is $l$-th linear order from $R_\Pi$. By McFadden and Richter [13] and KS, $\rho$ can be explained by a population of DMs whose preferences are in $R_\Pi$ if and only if

$$\rho = Bv$$

for some $v \in \mathbb{R}_+^{|R_\Pi|}$. This is our main result:

**Theorem 1.** *The following are equivalent:*

1. *A stochastic dataset $\rho$ admits an RRDEU representation.*
2. *A stochastic dataset $\rho$ is such that there exists some $v \in \mathbb{R}_+^{|R_\Pi|}$ such that $\rho = Bv$.*

This result generalizes some informal results about EU in Aguiar et al. [14]. The proof of Theorem 1 follows from Lemma 1, Proposition 1, and KS.

The RRDEU model is a strict generalization of REU. It is less general than RU because it predicts first-order stochastic dominance (under mild monotonicity constraints), which RU does not require. Quiggin [4] and Abdellaoui [3] provide additional implications of RDEU.

Now we show that in finite datasets, we can differentiate between REU and RRDEU by means of a stochastic version of the Allais paradox.

**Example 2** (Allais Paradox). *Consider $X = \{x_1 = 5M, x_2 = 1M, x_3 = 0\}$, where M denotes 1 million tokens, and four lotteries $p = (0, 1, 0)$, $q = (0.1, 0.89, 0.01)$, $r = (0, 0.11, 0.89)$, and $s = (0.1, 0, 0.9)$. Suppose that we observe that $\rho_{\{p,q\}}(p) \geq 0.5$ and $\rho_{\{r,s\}}(s) > 0.5$. This dataset cannot be consistent with REU. It has to be that if the DMs can be described by EU behavior, then $\rho_{\{r,s\}}(s) \leq 0.5$. The fact that $\rho_{\{p,q\}}(p) \geq 0.5$ means that more than half of the DMs that are consistent with EU should prefer r to s. In fact, this is a stochastic version of the Allais paradox. Thus, this dataset can be consistent with RRDEU as the Allais paradox can be accommodated by RDEU.*

### 3.3. Shape Restrictions

Our framework allows us to impose monotonicity or concavity/convexity on $\phi$. Monotonicity is a normatively desirable property. Abdellaoui [3] shows that convexity of $\phi$ is related to risk aversion. In particular, imposing it guarantees that consumers are risk-averse if the Bernoulli utility is set to the identity.

In particular, to impose the restriction that $\phi$ is weakly monotonically increasing, it suffices to enhance system (1) with the following set of linear inequality constraints:

$$f_{l,k} \geq f_{s,m}, \text{ whenever } \sum_{t=1}^{k} p_t^{\succ(l)} \geq \sum_{t=1}^{m} p_t^{\succ(s)}. \tag{2}$$

Convexity (concavity) can be imposed using cyclical monotonicity [15]. In particular, for all cycles of indices $\{l_j, k_j\}_{j=1}^{J}$ such that $l_J = l_1$ and $k_J = k_1$, let $\{f_{l,k}\}_{k=1,l=1}^{K-1,|\Pi|}$ satisfy

$$\sum_{j=1}^{J-1} \left( f_{l_{j+1},k_{j+1}} - f_{l_j,k_j} \right) \sum_{t=1}^{k_{j+1}} p_t^{\succ(l_{j+1})} \geq 0. \tag{3}$$

**Proposition 2.** *Given a finite set of lotteries $\Pi$, the linear order $\succ \in S$ is rank-dependent, expected utility linear order with weakly increasing and convex $\phi$ if and only if there exist $\{v_k\}_{k=1}^{K}$ and $\{f_{l,k}\}_{k=1,l=1}^{K-1,|\Pi|}$ such that system (1), together with restrictions (2) and (3), is satisfied.*

### 3.4. Econometric Testing

Here we deal with sampling variability. Sampling variability arises from the fact that $\rho$ can only be consistently estimated by the realized choice frequencies $\hat{\rho}$. This section follows closely with Aguiar et al. [16]. First, we need some additional notation. For every $A \in \mathcal{A}$, let $n_A$ denote the number of individuals in the sample that faced choice set $A$, and let $\mathbf{a}_{i,A}$, $i = 1, \ldots, n_A$, be the observed choice of individual $i$ from choice set $A$. Here we assume that the researcher observes a cross-section of observations for every $A \in \mathcal{A}$. Given this, we define the estimated stochastic choice rule as

$$\hat{\rho} = (\hat{\rho}_A(a))_{A \in \mathcal{A}, a \in A}.$$

with $\hat{\rho}_A(a) = n_A^{-1} \sum_{i=1}^{n_A} \mathbb{1}(\mathbf{a}_{i,A} = a)$ for any $a \in A$.

A natural test statistic based on Theorem 1 is

$$\mathrm{T}_n = n \min_{v \in \mathbb{R}_+^{|R_\Pi|}} \|\hat{\rho} - Bv\|^2,$$

where $n = \sum_A n_A$ is the sample size.

Let $\hat{\rho}_l^*$, $l = 1, \ldots, L$, be bootstrap replications of $\hat{\rho}$. Let $\tau_n \geq 0$ be a tuning parameter and $\iota$ be a vector of ones of dimension $|R_\Pi|$. (We conducted tests with $\tau_n = \sqrt{\log(\min_A n_A)/\min_A n_A}$ following KS.) To compute the critical values of $\mathrm{T}_n$, we follow the bootstrap procedure proposed in KS:

1.  Compute $\hat{\eta}_{\tau_n} = B v_{\tau_n}$, where $v_{\tau_n}$ solves

$$n \min_{[v - \tau_n \iota / |R_\Pi|] \in \mathbb{R}_+^{|R_\Pi|}} \| \hat{\rho} - Bv \|^2 ;$$

2.  Compute the bootstrap test statistic

$$\mathrm{T}_{n,l}^* = n \min_{[v - \tau_n \iota / d] \in \mathbb{R}_+^{|R_\Pi|}} \| \hat{\rho}_l^* - \hat{\rho} + \hat{\eta}_{\tau_n} - Bv \|^2, \quad l = 1, \dots, L;$$

3.  Use the empirical distribution of the bootstrap statistic to compute critical values of $\mathrm{T}_n$.

    For a given confidence level $\alpha \in (0, 1/2)$, the decision rule for the test is "reject the null hypothesis if $\mathrm{T}_n > \hat{c}_{1-\alpha}$," where $\hat{c}_{1-\alpha}$ is an $(1-\alpha)$-quantile of the empirical distribution of the bootstrap statistic.

## 4. Testing for RRDEU and REU in Experimental Data

To illustrate the proposed methodology, we use a subsample of the dataset from Aguiar et al. [16] to test for consistency of the behavior of a population of DMs with RRDEU and REU. Crucially, we use only one treatment (frame) of Aguiar et al. [16] corresponding to a situation where the cost of experimental subjects to pay attention is designed to be low. In that sense, we can avoid thinking about limited attention/consideration in this paper.

Aguiar et al. [16] conducted the experiment in Amazon MTurk for a large cross-section with at most two (disjoint) choice sets per individual. In the experiment, the set of prizes is $X = \{0, 10, 12, 20, 30, 48, 50\}$. The set of lotteries $\Pi$ is presented in Table 1. For example, the first lottery can be written as $l_1 = (1/2, 0, 0, 0, 0, 0, 1/2)'$ (i.e., it assigns positive probability to prizes 0 and 50 only). Similarly, the second lottery can be written as $l_2 = (0, 0, 1/2, 0, 0, 1/2, 0)'$.

All sessions were run between 25 August 2018 and 17 September 2018 on the MTurk platform with surveys designed in Qualtrics. The dataset contains 4099 choices from all possible non-singleton menus that contain the default lottery *o*, which pays 12 tokens with certainty. For more details on the experiment, see Aguiar et al. [16].

**Table 1.** Lotteries measured in tokens, expected values, and variance.

|      | **LOTTERY** | **EXPECTATION** | **VARIANCE** |
|------|-------------|-----------------|--------------|
| (1)  | $\frac{1}{2}50 + \frac{1}{2}0$ | 25 | 625 |
| (2)  | $\frac{1}{2}30 + \frac{1}{2}10$ | 20 | 100 |
| (3)  | $\frac{1}{4}50 + \frac{1}{4}30 + \frac{1}{4}10 + \frac{1}{4}0$ | 22.5 | 368.75 |
| (4)  | $\frac{1}{4}50 + \frac{1}{5}48 + \frac{3}{20}14 + \frac{2}{5}0$ | 24.125 | 511.73 |
| (5)  | $\frac{1}{5}48 + \frac{1}{4}30 + \frac{3}{20}14 + \frac{1}{4}10 + \frac{3}{20}0$ | 21.625 | 251.11 |
| (o)  | 12 with probability 1 | 12 | 0 |

**Structure of the Lotteries** Here, we show the special structure of the alternatives in the experimental dataset that allows us to test EU (and the independence axiom) as a restriction on the set of linear orders (i.e., we use $R_\Pi^{EU}$ to denote the restriction of the set of linear orders consistent with EU to $\Pi$). The experiment was design with power against REU. Here, we will use the Aguiar et al. [16] experiment to also test RRDEU.

If $\succ \in R_\Pi^{EU}$, then independence implies that for any $p, q, r \in \Delta(X)$ and any $\alpha \in (0, 1)$

$$p \succ q \iff \alpha p + (1 - \alpha) r \succ \alpha q + (1 - \alpha) r .$$

To understand additional restrictions imposed by independence, define the auxiliary lottery $r = (0, 2/5, 0, 3/10, 0, 0, 3/10)'$, and note the following relations among lotteries in $\Pi$:

$$l_3 = \frac{1}{2}l_1 + \frac{1}{2}l_2, \quad l_4 = \frac{1}{2}l_1 + \frac{1}{2}a, \quad l_5 = \frac{1}{2}l_2 + \frac{1}{2}a.$$

This structure restricts the possible orders that are compatible with expected utility: (i) if $l_1 \succ l_2$, then $l_1 \succ l_3, l_3 \succ l_2$, and $l_4 \succ l_5$; or (ii) if $l_2 \succ l_1$, then $l_2 \succ l_3, l_3 \succ l_1$, and $l_5 \succ l_4$.

However, the previous restrictions are only implications of the expected utility assumption. The necessary and sufficient conditions are spelled out in Theorem 1.

*Results*

We apply the procedure described in Section 3.4 to test whether REU and RRDEU can explain the data. The results of testing are presented in Table 2. In this table, we report the values of the test statistic and the corresponding *p*-values coming from the bootstrap distribution (1000 bootstrap replications for every test statistic were conducted) for two models. (The *p*-value is interpreted as the probability of observing a realization of the test statistic that is above the one that is actually observed due to sample variability, if the null hypothesis is indeed correct. Then, the smaller the *p*-value is, the more evidence there is that the researcher has to reject the hypothesis of the validity of a given model.) We reject the null hypothesis of expected utility at the 5-percent significance level (*p*-value = 0.013). At the same time, we cannot reject the rank-dependent expected utility model at any standard significance level (*p*-value = 0.906).

We must highlight that it is not entirely surprising that RRDEU is not rejected while REU is, since RRDEU is a strict generalization of REU. However, there is no reason a priori to think that RRDEU explains the experimental datasets used here. In that sense, we add some robust nonparametric empirical evidence supporting the use of RRDEU instead of REU to explain the choice over risky prospects. We must also mention that this is the first test of REU that takes into account sampling variability.

Other alternatives to EU such as Prospect Theory [5] may also be able to describe the experimental dataset we have used. Our current results cannot rule out other explanations. Evidently, Cumulative Prospect Theory [6] generalizes RDEU, so the results here support this model as well. However, an investigation of which alternative to EU is more successful is left for future work. In particular, our test has allowed the utility over money (i.e., Bernoulli utility) to be unconstrained. This means that if there is a fixed reference point, then we can cover loss aversion. Of course, taking into account reference points that change with menus is beyond the scope of our current investigation. (In particular, one would need an experimental design with power against violations of first-order stochastic dominance. This feature could allow differentiating between traditional probability weighting in the prospect theory versus RDEU.)

As we mentioned before, the experimental dataset we use was designed with power against REU. However, this dataset has rich menu variation. Thus, it has some power against all random utility rules (including RRDEU). We leave for future work the design of an experiment with additional power to test RRDEU and other popular extensions of EU.

We conclude this section by noting that RRDEU is one of the possible extensions of REU. REU may also fail if DMs do not consider all the lotteries in menus. That is, DMs may be prone to limited consideration [17]. (This concern is most likely not very important in our application, since we use the subsample from Aguiar et al. [16] that corresponds to low cost of attention.) To handle this type of violation of REU, the results in this paper can be coupled with the methodologies in Aguiar et al. [16] and Kashaev and Aguiar [18] to provide a test of RRDEU and REU with limited consideration.

**Table 2.** Testing Results.

| Model | $T_n$ | $p$-Value |
|---|---|---|
| REU | 387.72 | 0.013 |
| RRDEU | 130.75 | 0.906 |

Notes: Number of bootstrap replications = 1000.

## 5. Conclusions

We have proposed a new characterization of RDEU when we observe a cross-section of choices from heterogeneous DMs that choose from a finite set of lotteries and a finite collection of menus. We have established a nonparametric and computationally feasible test of RRDEU (and as a byproduct a test of REU). Our main result lends itself to statistical testing using the tools in KS. This test is widely applicable in experimental and field stochastic choice datasets.

We have obtained a characterization for REU that works for a finite collection of menus, which is not the case for the existing characterization by GP. We have shown that conditions characterizing REU in GP are not sufficient for a dataset defined on a finite collection of menus to be consistent with REU (Example 1). Using an experimental dataset collected by Aguiar et al. [16], we find evidence against REU and support for RRDEU.

We highlight that the techniques we have applied here to obtain the characterization of RRDEU could be used to provide analogous characterizations for other generalizations of EU such as Prospect Theory [5]. We leave for future work the exploration of the empirical content of other extensions of EU in a cross-section of choices. (The authors' names are in certified random order, as described by Ray and Robson [19]).

**Author Contributions:** Conceptualization, methodology, formal analysis, data curation, writing—original draft preparation, writing—review and editing: N.K. and V.H.A. Both authors have contributed equally to this manuscript. All authors have read and agreed to the published version of the manuscript.

**Funding:** This research received no external funding.

**Institutional Review Board Statement:** The experimental study we used in this research is approved by the Caltech's IRB No. 18-0812.

**Informed Consent Statement:** Informed consent was obtained from all subjects involved in the study.

**Data Availability Statement:** Restrictions apply to the availability of these data. Data was obtained from Victor H. Aguiar, Maria Boccardi, Jeongbin Kim, and Nail Kashaev and are available upon request from the authors with their permission.

**Acknowledgments:** We thank Maria Jose Boccardi and Jeongbin Kim for allowing the use of the experimental dataset collected jointly with the authors in previous projects. We are grateful to the three anonymous referees for useful comments.

**Conflicts of Interest:** The authors declare no conflict of interest.

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
