# Peer review of "Random Rank-Dependent Expected Utility"

_games, doi:10.3390/g13010013_

Round 1
Reviewer 1 Report
The paper “Random Rank-Dependent Expected Utility” is an interesting paper and provides new insights in utility theory. Globally it is well organized and well written. The fundamentals are described in a clear way.
The example explored could be more detailed, such as the interpretation of results. It would be good to have a deep discussion of these results in terms of implication to the traditional EU and also for other approaches, namely the Prospect Theory (considering the Allais Paradox).
The abstract is very poor, such as conclusions. The authors should describe better the main goal of this paper, such as the respective contribution.
Author Response
Dear Referee 1,
We are grateful for your careful reading of our work. We are also thankful for your positive assessment of our manuscript. Here we detail how we have addressed your comments/questions:
- We have expanded and clarified our Example 1. We have also added a second example (page 10) that presents a stochastic version of the Allais paradox and illustrates how our approach can differentiate between EU and RDEU.
- We have expanded our discussion of the results by relating our work with prospect theory and cumulative prospect theory (pages 15 and 17).
- We have fleshed out the abstract and the conclusions.
We hope you are satisfied with the new version of our manuscript.
Best,
The authors
Reviewer 2 Report
Summary and Assessment
Many axiomatic characterizations depend on rich domain assumptions. For example, if one is fundamentally limited to a finite data set, this may preclude identification of utility functions but it might also mean that characterizations (specifically, necessity of axioms) do not hold. The authors show that this is the case for Gul and Pesendorfer’s characterization of random expected utility.
The authors then look deeper for that particular model. For all models that are “random utility-like” extensions of nonstochastic models, a test on finite data is available by (1) listing the finitely many nonstochastic behaviors that are rationalizable under the nonstochastic model, (2) drawing on McFadden and Richter’s theoretical insights and Kitamura and Stoye’s operationalization to construct finite sample tests.
In the example, (1) can be done by repeatedly solving a relatively simple quadratic program (display (1)). All of this is executed by reanalyzing some data from the authors’ earlier work.
This is an interesting and useful exercise. The basic idea that the McFadden-Richter-(Kitamura-Stoye) machinery can be used for random extensions of different discrete choice models is not novel; it is clearly anticipated in those papers. But there is value in carrying this out for a well-known model and in clarifying (1). One might also remark that the characterization in (1) does not carry intellectual insight beyond, well, giving a criterion to determine if finite data are compatible with RDEU.
Questions to Authors
- RDEU was axiomatically characterized by Schmeidler (1989). Can you mention that and explain how display (1) relates to that characterization. Without revisiting it, I assume Schmeidler’s work has demanding domain conditions as well?
- Does the test have power against RDEU in your experimental setup?
Author Response
Dear Referee 2,
We thank you for your careful reading and for your positive assessment of our manuscript. Here we detail how we have addressed your comments/questions:
- We have added the reference to Schmeidler (1989) that characterized the nonadditive expected utility in environments with uncertainty. For the environment with risk we study, we reference the axiomatization of RDEU due to Abdellaoui (2002). We now specify in the introduction that the axiom, called “probability trade-off consistency,” replaces the independence axiom. This axiom requires, as you noted, a rich domain of lotteries that the typical finite collection of lotteries and menus does not have. See page 4.
- We clarify in our discussion of the results that the experimental dataset we use has power against random utility due to its rich menu variation. Since RRDEU is a special case of random utility the experimental dataset we use has power against RRDEU. Of course, there is additional power against REU due to the design of the lotteries, so we clarify this point and leave for future work the design of an experimental environment that is well suited to differentiate between different generalizations of REU. See page 15.
We hope you are satisfied with the new version of our manuscript.
Best,
The Authors.
Reviewer 3 Report
The Paper is interesting and well written; however, I have some concerns about the Introduction section. In the present version, the authors do not include a general overview of the problem or add some literature-related works. Moreover, they start with equations and examples. Therefore, I suggest starting with the general description of the problem, adding some references from literature, and then eventually moving to an example.
Some minor remarks:
- The header "Data" in Section 4 can be omitted. The sentence "The data contains choices 4099 choices (...)" at the bottom of page 12 has double "choices" word.
- In the "Results" section, I would use p=0.05 instead of "5 percent significance level".
- The "Conclusions" section is very concise. However, authors should extend it by a short description of future works.
Author Response
Dear Referee 3,
We thank you for your careful reading and for your positive assessment of our manuscript. Here we detail how we have addressed your comments/questions:
- We have fleshed out the descriptive part of our introduction. We have added additional relevant references and connected with the literature in a better way. Evidently, there is a huge number of references related to RDEU so we have cited relevant survey papers. We must mention that we did not change entirely the style of our introduction since this is a very short revision (we were given 5 days). We hope you agree this is a matter of style, and that this paper is targeted to specialists.
- Thanks for your suggestions regarding the labelling of subsections and finding a We have changed our labels and also we have corrected this typo.
- Now we have added the p-values to our text as you recommend.
- We have fleshed out the conclusion and we now discuss there the directions for future work.
We hope you are satisfied with the new version of our manuscript.
Best,
The Authors